# Parametrized Hierarchical Procedures for Neural Programming

**Roy Fox, Richard Shin, Sanjay Krishnan, Ken Goldberg, Dawn Song, and Ion Stoica**
Department of Electrical Engineering and Computer Sciences
University of California, Berkeley
`{royf,ricshin,sanjaykrishnan,goldberg,dawnsong,istoica}@berkeley.edu`

## Abstract

Neural programs are highly accurate and structured policies that perform algorithmic tasks by controlling the behavior of a computation mechanism. Despite the potential to increase the interpretability and the compositionality of the behavior of artificial agents, it remains difficult to learn from demonstrations neural networks that represent computer programs. The main challenges that set algorithmic domains apart from other imitation learning domains are the need for high accuracy, the involvement of specific structures of data, and the extremely limited observability. To address these challenges, we propose to model programs as Parametrized Hierarchical Procedures (PHPs). A PHP is a sequence of conditional operations, using a program counter along with the observation to select between taking an elementary action, invoking another PHP as a sub-procedure, and returning to the caller. We develop an algorithm for training PHPs from a set of supervisor demonstrations, only some of which are annotated with the internal call structure, and apply it to efficient level-wise training of multi-level PHPs. We show in two benchmarks, NanoCraft and long-hand addition, that PHPs can learn neural programs more accurately from smaller amounts of both annotated and unannotated demonstrations.

## 1 Introduction

Representing the logic of a computer program with a parametrized model, such as a neural network, is a central challenge in AI with applications including reinforcement learning, robotics, natural language processing, and programming by example. A salient feature of recently-proposed approaches for learning programs (Reed & De Freitas, 2016; Cai et al., 2017; Li et al., 2017) is their ability to leverage the hierarchical structure of procedure invocations present in well-designed programs.

Explicitly exposing this hierarchical structure enables learning neural programs with empirically superior generalization, compared to baseline methods that learn only from elementary computer operations, but requires training data that does not consists only of low-level computer operations but is annotated with the higher-level procedure calls (Reed & De Freitas, 2016; Cai et al., 2017). Li et al. (2017) tackled the problem of learning hierarchical neural programs from a mixture of annotated training data (hereafter called *strong supervision*) and unannotated training data where only the elementary operations are given without their call-stack annotations (called *weak supervision*). In this paper, we propose to learn hierarchical neural programs from a mixture of strongly supervised and weakly supervised data via the Expectation–Gradient method and an explicit program counter, in lieu of a high-dimensional real-valued state of a recurrent neural network.

Our approach is inspired by recent work in robot learning and control. In Imitation Learning (IL), an agent learns to behave in its environment using supervisor demonstrations of the intended behavior. However, existing approaches to IL are largely insufficient for addressing *algorithmic domains*, in which the target policy is program-like in its accurate and structured manipulation of inputs and data structures. An example of such a domain is long-hand addition, where the computer loops over the digits to be added, from least to most significant, calculating the sum and carry. In more complicated examples, the agent must correctly manipulate data structures to compute the right output.

Three main challenges set algorithmic domains apart from other IL domains. First, the agent's policy must be highly accurate. Algorithmic behavior is characterized by a hard constraint of output correctness, where any suboptimal actions are simply wrong and considered failures. In contrast, many tasks in physical and simulated domains tolerate errors in the agent's actions, as long as some goal region in state-space is eventually reached, or some safety constraints are satisfied. A second challenge is that algorithms often use specific data structures, which may require the algorithmic policies to have a particular structure. A third challenge is that the environment in algorithmic domains, which consists of the program input and the data structures, is almost completely unobservable directly by the agent. They can only be scanned using some limited reading apparatus, such as the read/write heads in a Turing Machine or the registers in a register machine.

Recently proposed methods can infer from demonstration data hierarchical control policies, where high-level behaviors are composed of low-level manipulation primitives (Daniel et al., 2016; Fox et al., 2017). In this paper, we take a similar approach to address the challenges of algorithmic domains, by introducing *Parametrized Hierarchical Procedures* (PHPs), a structured model of algorithmic policies inspired by the *options* framework (Sutton et al., 1999), as well as the *procedural programming* paradigm. A PHP is a sequence of statements, such that each statement branches conditionally on the observation, to either (1) perform an elementary operation, (2) invoke another PHP as a sub-procedure, or (3) terminate and return control to the caller PHP. The index of each statement in the sequence serves as a *program counter* to accurately remember which statement was last executed and which one is next. The conditional branching in each statement is implemented by a neural network mapping the program counter and the agent's observation into the elementary operation, sub-procedure, or termination to be executed. The PHP model is detailed in Section 4.1.

PHPs have the potential to address the challenges of algorithmic domains by strictly maintaining two internal structures: a call stack containing the current branch of caller PHPs, and the current program counter of each PHP in the stack. When a statement invokes a PHP as a sub-procedure, this PHP is pushed into the call stack. When a statement terminates the current PHP, it is popped from the stack, returning control to the calling PHP to execute its next statement (or, in the case of the root PHP, ending the entire episode). The stack also keeps the program counter of each PHP, which starts at 0, and is incremented each time a non-terminating statement is executed.

PHPs impose a constraining structure on the learned policies. The call stack arranges the policy into a hierarchical structure, where a higher-level PHP can solve a task by invoking lower-level PHPs that solve sub-tasks. Since call stacks and program counters are widely useful in computer programs, they provide a strong inductive bias towards policy correctness in domains that conform to these constraints, while also being computationally tractable to learn. To support a larger variety of algorithmic domains, PHPs should be extended in future work to more expressive structures, for example allowing procedures to take arguments.

We experiment with PHPs in two benchmarks, the NanoCraft domain introduced in Li et al. (2017), and long-hand addition. We find that our algorithm is able to learn PHPs from a mixture of strongly and weakly supervised demonstrations with better sample complexity than previous algorithms: it achieves better test performance with fewer demonstrations.

In this paper we make three main contributions:

- We introduce the PHP model and show that it is easier to learn than the NPI model (Reed & De Freitas, 2016).
- We propose an Expectation–Gradient algorithm for efficiently training PHPs from a mixture of annotated and unannotated demonstrations (strong and weak supervision).
- We demonstrate efficient training of multi-level PHPs on NanoCraft (Li et al., 2017) and long-hand addition (Reed & De Freitas, 2016), and achieve improved success rate.

## 2 RELATED WORK

### 2.1 NEURAL PROGRAMMING

Using input–output examples to specify a task has been a common setting for learning programs with neural networks. Various architectures, such as the Neural Turing Machine (Graves et al.,

| | Execution traces | | Task specification format | |
|---|---|---|---|---|
| System | Low-level actions | Higher-level structure | Input–output pairs | Natural language |
| Graves et al. (2014) Sukhbaatar et al. (2015) Kaiser & Sutskever (2015) Joulin & Mikolov (2015) | ✗ | ✗ | ✓ | ✗ |
| Neelakantan et al. (2016) Andreas et al. (2016) | ✗ | ✗ | ✗ | ✓ |
| Andreas et al. (2017) | ✗ | ✓ | ✗ | ✓ |
| NPI (Reed & De Freitas, 2016) | ✓ | ✓ | ✓ | ✗ |
| Recursive NPI (Cai et al., 2017) | ✓ | ✓ (recursive) | ✓ | ✗ |
| NPL (Li et al., 2017) | ✓ | Mixed | ✓ | ✗ |
| **PHP (this work)** | ✓ | Mixed | ✓ | ✗ |

Table 1: Summary of related work in neural programming. Each column indicates what data is used by the system. "Mixed": only some of the training data is annotated with the higher-level structure.

2014), Stack RNNs (Joulin & Mikolov, 2015), the Neural GPU (Kaiser & Sutskever, 2015), and End-to-End Memory Networks (Sukhbaatar et al., 2015), have been proposed for learning neural programs from input–output examples, with components such as variable-sized memory and novel addressing mechanisms facilitating the training process.

In contrast, our work considers the setting where, along with the input–output examples, execution traces are available which describe the steps necessary to solve a given problem. The Neural Programmer–Interpreter (NPI, Reed & De Freitas (2016)) learns hierarchical policies from execution traces which not only indicate the low-level actions to perform, but also a structure over them specified by higher-level abstractions. Cai et al. (2017) showed that learning from an execution trace with recursive structure enables perfect generalization. Neural Program Lattices (Li et al., 2017) work within the same setting as the NPI, but can learn from a dataset of execution traces where only a small fraction contains information about the higher-level hierarchy.

In demonstrations where the hierarchical structure along the trace is missing, this latent space grows exponentially in the trace length. Li et al. (2017) address this challenge via an approximation method that selectively averages latent variables on different computation paths to reduce the complexity of enumerating all paths. In contrast, we compute exact gradients using dynamic programming, by considering a hierarchical structure that has small discrete latent variables in each time step.

Other works use neural networks as a tool for outputting programs written in a discrete programming language, rather than having the neural network itself represent a program. Balog et al. (2017) learned to generate programs for solving competition-style problems. Devlin et al. (2017) and Parisotto et al. (2017) generate programs in a domain-specific language for manipulating strings in spreadsheets.

## 2.2 HIERARCHICAL CONTROL

Automatic discovery of hierarchical structure has been well-studied, and successful approaches include action-sequence compression (Thrun & Schwartz, 1994), identifying important transitional states (McGovern & Barto, 2001; Menache et al., 2002; Şimşek & Barto, 2004; Stolle, 2004; Lakshminarayanan et al., 2016), learning from demonstrations (Bui et al., 2002; Krishnan et al., 2015; Daniel et al., 2012; Krishnan et al., 2016), considering the set of initial states from which the MDP can be solved (Konidaris & Barto, 2009; Konidaris et al., 2012), policy gradients (Levy & Shimkin, 2011), information-theoretic considerations (Genewein et al., 2015; Fox et al., 2016; Jonsson & Gómez, 2016; Florensa et al., 2017), active learning (Hamidi et al., 2015), and recently value-function approximation (Bacon et al., 2017; Heess et al., 2016; Sharma et al., 2017).

Our approach is inspired by the *Discovery of Deep Options (DDO)* algorithm of Fox et al. (2017). Following the work of Daniel et al. (2016), who use Expectation–Maximization (EM) to train an Abstract Hidden Markov Model (Bui et al., 2002), DDO parametrizes the model with neural networks where complete maximization in the M-step is infeasible. Instead, DDO uses Expectation–Gradient (EG) to take a single gradient step using the same forward–backward E-step as in the EM algorithm. A variant of DDO for continuous action spaces (DDCO) has shown success in simulated and physical robot control (Krishnan et al., 2017). This paper extends DDO by proposing an E-step that can infer a call-stack of procedures and their program counters.

## 3 PROBLEM STATEMENT

Computation can be modeled as a deterministic dynamical system, where the computer is an agent interacting with its environment, which consists of the program input and its data structures. Mathematically, the environment is a Deterministic Partially Observable Markov Decision Process (DET-POMDP (Bonet, 2009)), which consists of a state space $\mathcal{S}$, an observation space $\mathcal{O}$, an action space $\mathcal{A}$, the state-dependent observation $o_t(s_t)$, and the state transition $s_{t+1} = f(s_t, a_t)$. The initial state $s_0$ includes the program input, and is generated by some distribution $p_0(s_0)$. This notation is general enough to model various computation processes. In a Turing Machine, for example, $s_t$ is the machine's configuration, $o_t$ is the vector of tape symbols under the read/write heads, and $a_t$ contains writing and moving instructions for the heads.

In partially observable environments, the agent often benefits from maintaining memory $m_t$ of past observations, which reveals temporarily hidden aspects of the current state. The agent has a parametrized stochastic policy $\pi_\theta$, in some parametric family $\theta \in \Theta$, where $\pi_\theta(m_t, a_t | m_{t-1}, o_t)$ is the probability of updating the memory state from $m_{t-1}$ to $m_t$ and taking action $a_t$, when the observation is $o_t$. The policy can be rolled out to induce the stochastic process $(s_{0:T}, o_{0:T}, m_{0:T-1}, a_{0:T-1})$, such that upon observing $o_T$ the agent chooses to terminate the process. In a computation device, the memory $m_t$ stands for its internal state, such as the Finite State Machine of a Turing Machine. We can scale computer programs by adding data structures to their internal state, such as a call stack, which we model in the next section as Parametrized Hierarchical Procedures.

In Imitation Learning (IL), the learner is provided with direct supervision of the correct actions to take. The setting we use is Behavior Cloning (BC), where the supervisor rolls out its policy to generate a batch of demonstrations before learning begins, and the agent's policy is trained to minimize a loss on its own selection of actions in demonstrated states, with respect to the demonstrated actions. In *strong supervision*, a demonstration contains not only the sequence of observable variables $\xi = (o_{0:T}, a_{0:T-1})$, where $a_{0:T-1}$ is the sequence of supervisor actions during the demonstration, but also the sequence of the supervisor's memory states $\zeta = m_{0:T-1}$, which are ordinarily latent. This allows the agent to directly imitate not just the actions, but also the memory updates of the supervisor, for example by maximizing the log-likelihood of the policy given the demonstrations

$$\arg\max_\theta \sum_i \log \mathbb{P}(\zeta_i, \xi_i | \theta) = \arg\max_\theta \sum_i \sum_{t=0}^{T_i-1} \log \pi_\theta(m_{i,t}, a_{i,t} | m_{i,t-1}, o_{i,t}),$$

the latter being the negative cross-entropy loss with respect to the demonstrations.

In *weak supervision*, on the other hand, only the observable trajectories $\xi$ are given as demonstrations. This makes it difficult to maximize the likelihood $\mathbb{P}(\xi | \theta) = \sum_\zeta \mathbb{P}(\zeta, \xi | \theta)$, due to the large space of possible memory trajectories $\zeta$. We address this difficulty via the Expectation–Gradient algorithm described in Section 4.2.

## 4 PARAMETRIZED HIERARCHICAL PROCEDURES

### 4.1 DEFINITION

#### 4.1.1 HIERARCHICAL PROCEDURES

A finite set $\mathcal{H}$ of *hierarchical procedures* can be defined recursively as follows. Each hierarchical procedure $h \in \mathcal{H}$ is a sequence $\sigma_h^0, \sigma_h^1, \ldots$ of *statements*. A statement $\sigma_h^\tau = (\eta_h^\tau, \psi_h^\tau)$ consists of an

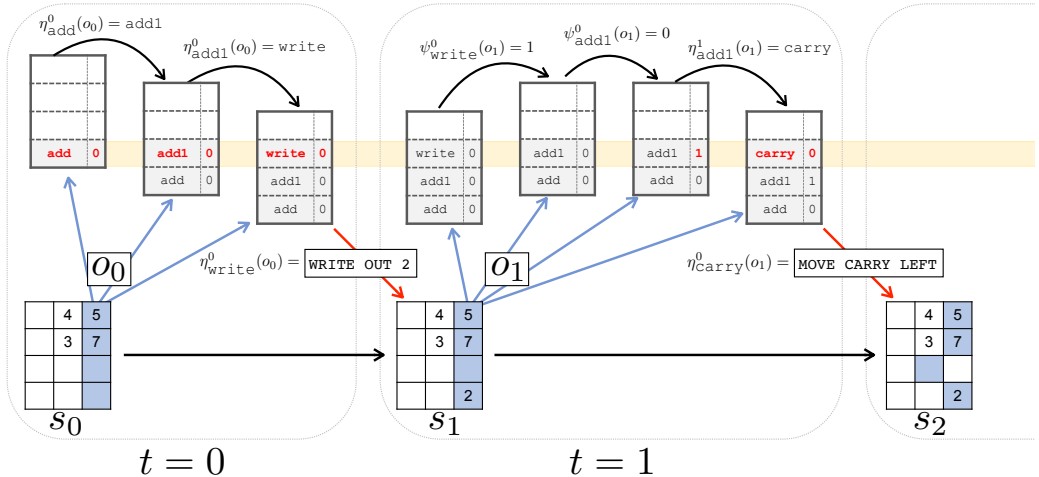

Figure 1: Execution of a Parametrized Hierarchical Procedure (PHP) on the long-hand addition domain. The highest-level `add` procedure invokes a lower-level procedure `add1` which further invokes `write`. `write` takes the elementary action `WRITE OUT 2`, and returns to `add1`. `add1` chooses not to return, incrementing its program counter to 1, and then invoking `carry`, which in turn takes the elementary action `MOVE CARRY LEFT`.

*operation statement* $\eta_h^\tau$ and a *termination statement* $\psi_h^\tau$. The operation statement $\eta_h^\tau : \mathcal{O} \to \mathcal{A} \cup \mathcal{H}$ is a conditional branching block that selects at step $\tau$ of procedure $h$, based on the external observation, either an elementary action to execute or another hierarchical procedure to invoke. The termination statement $\psi_h^\tau : \mathcal{O} \to \{0, 1\}$ is a conditional termination indicator that decides, based on the external observation, whether to terminate the procedure $h$ after step $\tau$. One of the procedures is the *root* of the hierarchy.

The semantics of this definition are given by the following control policy. The agent's memory maintains a stack $[(h_1, \tau_1), \ldots, (h_n, \tau_n)]$ of the active procedures and their program counters. Initially, this stack contains only the root procedure and the counter is 0. Upon observing $o_t$, the agent checks whether the top procedure should terminate, i.e. $\psi_{h_n}^{\tau_n}(o_t) = 1$. If the procedure $h_n$ terminates, it is popped from the stack, the next termination condition $\psi_{h_{n-1}}^{\tau_{n-1}}(o_t)$ is consulted, and so on. For the first procedure $h_i$ that does not terminate, we select the operation $\eta_{h_i}^{\tau_i+1}(o_t)$, after incrementing the program counter $\tau_i$. If this operation is an invocation of procedure $h_{i+1}'$, we push $(h_{i+1}', 0)$ onto the stack, consult its operation statement $\eta_{h_{i+1}'}^0(o_t)$, and so on. Upon the first procedure $h_{n'}'$ to select an elementary action $a_t$, we save the new memory state $m_t = [(h_1, \tau_1), \ldots, (h_{i-1}, \tau_{i-1}), (h_i, \tau_i + 1), (h_{i+1}', 0), \ldots, (h_{n'}', 0)]$, and take the action $a_t$ in the environment.

The call stack and program counters act as memory for the agent, so that it can remember certain hidden aspects of the state that were observed before. In principle, any finite memory structure can be implemented with sufficiently many PHPs, by having a distinct procedure for each memory state. However, PHPs leverage the call stack and program counters to allow exponentially many memory states to be expressed with a relatively small set of PHPs.

We impose two practical limitations on the general definition of PHPs. Our training algorithm in Section 4.2 does not support recursive procedures, i.e. cycles in the invocation graph. In addition, for simplicity, we allow each procedure to either invoke other procedures or execute elementary actions, not both. These two limitations are achieved by layering the procedures in levels, such that only the lowest-level procedures can execute elementary actions, and each higher-level procedure can

only invoke procedures in the level directly below it. This does not lose generality, since instead of invoking a procedure or action at a certain level, we can wrap it in a one-level-higher surrogate procedure that invokes it and terminates.

### 4.1.2 PARAMETRIZED HIERARCHICAL PROCEDURES

A Parametrized Hierarchical Procedure (PHP) is a representation of a hierarchical procedure by differentiable parametrization. In this paper, we represent each PHP by two multi-layer perceptrons (MLPs) with ReLU activations, one for the PHP's operation statement and one for its termination statement. The input is a concatenation of the observation $o$ and the program counter $\tau$, where $\tau$ is provided to the MLPs as a real number. During training, we apply the soft-argmax activation function to the output of each MLP to obtain stochastic statements $\eta_h^\tau(\cdot|o_t)$ and $\psi_h^\tau(\cdot|o_t)$. During testing, we replace the soft-argmax with argmax, to obtain deterministic statements as above.

### 4.2 TRAINING ALGORITHM

### 4.2.1 EXPECTATION–GRADIENT METHOD

In weak supervision, only the observable trajectory $\xi = (o_{0:T}, a_{0:T-1})$ is available in a demonstration, and the sequence of memory states $\zeta = m_{0:T-1}$ is latent. This poses a challenge, since the space of possible memory trajectories $\zeta$ grows exponentially in the length of the demonstration, which at first seems to prohibit the computation of the log-likelihood gradient $\nabla_\theta \log \mathbb{P}(\xi|\pi_\theta)$, needed to maximize the log-likelihood via gradient ascent.

We use the Expectation–Gradient (EG) method to overcome this challenge (Salakhutdinov et al., 2003). This method has been previously used in dynamical settings to play Atari games (Fox et al., 2017) and to control simulated and physical robots (Krishnan et al., 2017). The EG trick expresses the gradient of the observable log-likelihood as the expected gradient of the full log-likelihood:

$$\nabla_\theta \log \mathbb{P}(\xi|\theta) = \frac{1}{\mathbb{P}(\xi|\theta)} \nabla_\theta \mathbb{P}(\xi|\theta) = \sum_\zeta \frac{1}{\mathbb{P}(\xi|\theta)} \nabla_\theta \mathbb{P}(\zeta, \xi|\theta)$$

$$= \sum_\zeta \frac{\mathbb{P}(\zeta, \xi|\theta)}{\mathbb{P}(\xi|\theta)} \nabla_\theta \log \mathbb{P}(\zeta, \xi|\theta) = \mathbb{E}_{\zeta|\xi,\theta}[\nabla_\theta \log \mathbb{P}(\zeta, \xi|\theta)],$$

where the first and third equations follow from two applications of the identity $\nabla_\theta x = x \nabla_\theta \log x$. In the E-step of the EG algorithm, we find the posterior distribution of $\zeta$ given the observed $\xi$ and the current parameter $\theta$. In the G-step, we use this posterior to calculate and apply the exact gradient of the observable log-likelihood.

### 4.2.2 TRAINING TWO-LEVEL PHPs

We start by assuming a shallow hierarchy, where the root PHP calls level-one PHPs that only perform elementary operations. At any time $t$, the stack contains two PHPs, the root PHP and the PHP it invoked to select the elementary action. The stack also contains the program counters of these two PHPs, however we ignore the root counter to reduce complexity, and bring it back when we discuss multi-level hierarchies in the next section.

Let us denote by $\eta_h^\tau(a_t|o_t)$ and $\psi_h^\tau(b_t|o_t)$, respectively, the stochastic operation and termination statements of procedure $h \in \mathcal{H} \cup \{\perp\}$, where $\perp$ is the root PHP. Let $(h_t, \tau_t)$ be the top stack frame when action $a_t$ is selected. Then the full likelihood $\mathbb{P}(\zeta, \xi|\theta)$ of the policy given an annotated demonstration is a product of the terms that generate the demonstration, including $\eta_{h_t}^{\tau_t}(a_t|o_t)$ for the generation of each $a_t$, as well as $\psi_{h_{t-1}}^{\tau_{t-1}}(1|o_t)\eta_\perp(h_t|o_t)$ whenever $h_{t-1}$ terminates and $h_t$ is pushed with $\tau_t = 0$, and $\psi_{h_{t-1}}^{\tau_{t-1}}(0|o_t)$ whenever $h_{t-1}$ does not terminate (i.e. $h_t = h_{t-1}$ and $\tau_t = \tau_{t-1}+1$).

Crucially, the form of $\mathbb{P}(\zeta, \xi|\theta)$ as a product implies that $\nabla_\theta \log \mathbb{P}(\zeta, \xi|\theta)$ decomposes into a sum of policy-gradient terms such as $\nabla_\theta \log \eta_{h_t}^{\tau_t}(a_t|o_t)$, and computing its expectation over $\mathbb{P}(\zeta|\xi, \theta)$ only requires the marginal posterior distributions over single-step latent variables

$$v_t(h, \tau) = \mathbb{P}(h_t{=}h, \tau_t{=}\tau|\xi, \theta)$$
$$w_t(h, \tau) = \mathbb{P}(h_t{=}h, \tau_t{=}\tau, \tau_{t+1}{=}\tau{+}1|\xi, \theta).$$

The marginal posteriors $v_t$ and $w_t$ can be found via a forward–backward algorithm, as described in Appendix A, and used to compute the exact gradient

$$\nabla_\theta \mathbb{P}(\xi|\theta) = \sum_{h \in \mathcal{H}} \sum_{t=0}^{T-1} \Bigg( v_t(h,0) \nabla_\theta \log \eta_\perp(h|o_t)$$
$$+ \sum_{\tau=0}^{t} \Bigg( v_t(h,\tau) \nabla_\theta \log \eta_h^\tau(a_t|o_t)$$
$$+ w_t(h,\tau) \nabla_\theta \log \psi_h^\tau(0|o_{t+1}))$$
$$+ (v_t(h,\tau) - w_t(h,\tau)) \nabla_\theta \log \psi_h^\tau(1|o_{t+1}) \Bigg) \Bigg).$$

### 4.2.3 TRAINING MULTI-LEVEL PHPS

A naive attempt to generalize the same approach to multi-level PHPs would result in an exponential blow-up of the forward–backward state, which would need to include the entire stack. Instead, we train each level separately, iterating over the PHP hierarchy from the lowest level to the highest. Let us denote by $d$ the number of levels in the hierarchy, with 0 being the root and $d-1$ the lowest level, then we train level $i$ in the hierarchy after we have trained levels $i+1, \ldots, d-1$.

Two components are required to allow this separation. First, we need to use our trained levels $i+1, \ldots, d-1$ to abstract away from the elementary actions, and generate demonstrations where the level-$(i+1)$ PHPs are treated as the new elementary operations. In this way, we can view level-$i$ PHPs as the new lowest-level PHPs, whose operations are elementary in the demonstrations. This is easy to do in strongly supervised demonstrations, since we have the complete stack, and we only need to truncate the lowest $d-i-1$ levels. In weakly supervised demonstrations, on the other hand, we need an algorithm for decoding the observable trajectories, and replacing the elementary actions with higher-level operations. We present such an algorithm below.

The second component needed for level-wise training is approximate separation from higher levels that have not been trained yet. When we train level $i > 1$ via the EG algorithm in the previous section, the "root PHP" would be at level $i-1$, had it corresponded to any real PHP. In all but the simplest domains, we cannot expect a single PHP to perfectly match the behavior of the $i$-levels PHP hierarchy (levels $0, \ldots, i-1$) that actually selected the level-$i$ PHPs that generated the demonstrations. To facilitate better separation from higher levels, we augment the "root PHP" used for training with an LSTM that approximates the $i$-levels stack memory as $\eta_\perp^{LSTM}(h_t|o_0, \ldots, o_t)$.

As mentioned above, abstraction from lower levels is achieved by rewriting weakly supervised demonstrations to show level-$(i+1)$ operations as elementary. After level $i+1$ is trained, the level-$(i+1)$ PHPs that generated the demonstrations are decoded using the trained parameters. We considered three different decoding algorithms: (1) finding the most likely level-$(i+1)$ PHP at each time step, by taking the maximum over $v_t$; (2) using a Viterbi-like algorithm to find the most likely latent trajectory of level-$(i+1)$ PHPs; (3) sampling from the posterior distribution $\mathbb{P}(\zeta|\xi,\theta)$ over latent trajectories. In our current experiments we used latent trajectories sampled from the posterior distribution, given by

$$\mathbb{P}(\zeta|\xi,\theta) = v_0(h_0,\tau_0) \prod_{t=0}^{T-2} \frac{\mathbb{P}(h_t,\tau_t,h_{t+1},\tau_{t+1}|\xi,\theta)}{v_t(h_t,\tau_t)},$$

where the denominators can be computed via the same forward–backward algorithm used in the previous section to compute $v_t$ and $w_t$, as detailed in Appendix A.

## 5 EXPERIMENTS

We evaluate our proposed method on the two settings studied by Li et al. (2017): NanoCraft, which involves an agent interacting in a grid world, and long-hand addition, which was also considered by Reed & De Freitas (2016) and Cai et al. (2017).

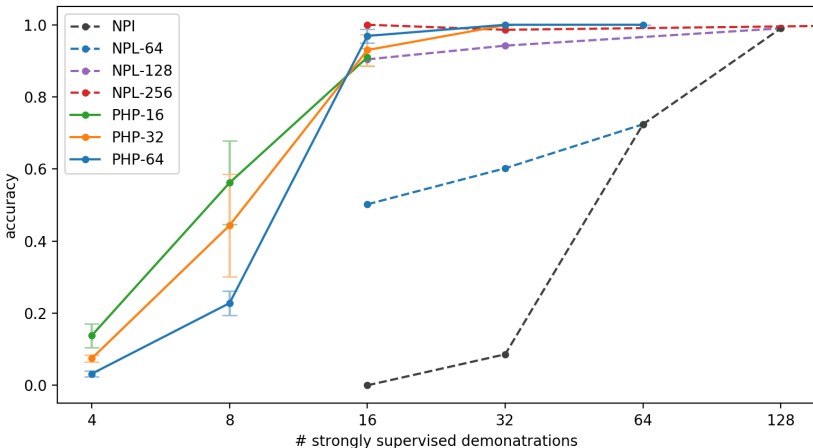

Figure 2: Sample complexity in the NanoCraft domain. The accuracy is the fraction of completely correct test episodes, as a function of the number of demonstrations annotated with the supervisor's hierarchy. PHP-{16, 32, 64} shows our results for training PHPs from the indicated total number of demonstrations. The results for NPL-{64, 128, 256} and NPI were provided by Li et al. (2017).

## 5.1 NANOCRAFT

**Task description.** The NanoCraft domain, introduced by Li et al. (2017), involves placing blocks in a two-dimensional grid world. The goal of the task is to control an agent to build a rectangular building of a particular height and width, at a specified location within the grid, by moving around the grid and placing blocks in appropriate cells.

The state contains a $6 \times 6$ grid. In our version, each grid cell can either be empty or contain a block. The state also includes the current location of the agent, as well as the building's desired height, width, and location, expressed as the offset from the agent's initial location at the top left corner. Initially, some of the blocks are already in place and must not be placed again.

The state-dependent observation $o_t(s_t)$ reveals whether the grid cell at which the agent is located contains a block or not, and four numbers for the building's specifications. We provide each observation to the MLPs as a 5-dimensional real-valued feature vector.

**PHPs and elementary actions.** The top-level PHP `nanocraft` executes (`moves_r`, `moves_d`, `builds_r`, `builds_d`, `builds_l`, `builds_u`, `return`). `moves_r` calls `move_r` a number of times equal to the building's horizontal location, and similarly for `moves_d` w.r.t. `move_d` and the vertical location; `builds_r` w.r.t. `build_r` and the building's width; and so on for `builds_d`, `builds_l`, and `builds_u`. At the lowest level, `move_r` takes the elementary action `MOVE_RIGHT` and terminates, and similarly for `move_d` taking `MOVE_DOWN`. `build_r` executes (`MOVE_RIGHT, if cell full:  return, else:  PLACE_BLOCK, return`), and similarly for `build_d`, `build_l`, and `build_u` w.r.t. `MOVE_DOWN`, `MOVE_LEFT`, and `MOVE_UP`.

**Experiment setup.** We trained our model on datasets of 16, 32, and 64 demonstrations, of which some are strongly supervised and the rest weakly supervised. We trained each level for 2000 iterations, iteratively from the lowest level to the highest. The results are averaged over 5 trials with independent datasets.

**Results.** Our results summarized in Figure 2 show that 32 strongly supervised demonstrations are sufficient for achieving perfect performance at the task, and that 16 such demonstrations approach the same success rate when used along with weakly supervised demonstrations, for a total of 16, 32, or 64 demonstrations.

An interesting question is whether these performance gains are due to the simplicity of the PHP model itself, the use of exact gradients in its optimization via the EG algorithm, or both. The PHP

and NPL/NPI experiments with 64 strongly supervised demonstrations (Figure 2, blue curves at the 64 mark) directly compare the PHP model with the NPI model, since both algorithms use exact gradients in this case.[1] The accuracy is 1.0 for PHP; 0.724 for NPL/NPI, suggesting that the gains of PHP are at least in part due to the PHP model inducing an optimization landscape in which a good solution is easier to find. In the experiments with 16 strongly supervised demonstrations of a total 64 (blue curves at the 16 mark), the success rate is 0.969 for PHP; 0.502 for NPL. This 70% increase in the gain of PHP over NPL may be evidence that exact gradients are better at training the model than the approximate gradients of NPL, although the choice of an optimization method is conflated here with the choice of a model.

## 5.2 LONG-HAND ADDITION

**Task description.** The long-hand addition task was also considered by Reed & De Freitas (2016), Li et al. (2017), and Cai et al. (2017). In this task, our goal is to add two numbers represented in decimal, by starting at the rightmost column (least significant digit) and repeatedly summing each column to write the resulting digit and a carry if needed. The state consists of 4 tapes, as in a Turing Machine, corresponding to the first number, the second number, the carries, and the output. The state also includes the positions of 4 read/write heads, one for each tape. Initially, each of the first two tapes contains the $K$ digits of a number to be added, all other cells contain the empty symbol, and the heads point to the least significant digits.

The state-dependent observation $o_t(s_t)$ reveals the value of the digits (or empty symbols) pointed to by the pointers. The four values are provided to the MLPs in one-hot encoding, i.e., the input vector has $11 \times 4$ dimensions with exactly one 1-valued entry in each of the four group.

**PHPs and elementary actions.** The top-level PHP `add` repeatedly calls `add1` to add each column of digits. `add1` calls `write`, `carry`, and `lshift` in order to compute the sum of the column, write the carry in the next column, and move the pointers to the next column. If the sum for a column is less than 10, then `add1` does not call `carry`.

There are two kinds of elementary actions: one which moves a specified pointer in a specified direction (e.g. `MOVE CARRY LEFT`), and one which writes a specified digit to a specified tape (e.g. `WRITE OUT 2`). $\eta_{\text{write}}, \eta_{\text{carry}}$, and $\eta_{\text{lshift}}$ output the probability distribution over possible action and argument combinations as the product of 3 multinomial distributions, each with 2, 4, and 10 possibilities respectively.

**Experiment setup.** Following Li et al. (2017), we trained our model on execution traces for inputs of each length 1 to 10. We used 16 traces for each input length, for a total of 160 traces.[2] We experimented with providing 1, 2, 3, 5, and 10 strongly supervised traces, with the remainder containing only the elementary actions.

For training our model, we performed a search over two hyperparameters:

- Weight on loss from strongly supervised traces: When the number of weakly supervised demonstrations overwhelms the number of strongly supervised traces, the model can learn a hierarchy which does not match the supervisor. By appropriately scaling up the loss contribution from the strongly supervised traces, we can ensure that the model learns to follow the hierarchy specified in them.
- Use of $\tau$ in $\psi$: The termination condition $\psi_h^\tau(b_t|o_t)$ contains a dependence on $\tau$, the number of steps that the current procedure $h$ has executed. However, sometimes the underlying definition for $\psi$ does not contain any dependence on $\tau$: $\psi_h^1(b|o) = \psi_h^2(b|o) = \cdots$. In such a case, the MLP for $\psi_h$ may learn a spurious dependency on $\tau$, and generalize poorly to values of $\tau$ seen during test time. Therefore, we searched over whether to use $\tau$ for $\psi$ at each level of the hierarchy.

**Results.** Our results are summarized in Table 2. Previous work by Li et al. (2017) learns a model which can generalize perfectly to input lengths of 500 but not 1000. In our experiments with the same

---

[1] Li et al. (2017) used batch size 1, whereas we used full batches and made no attempt to optimize batch size.
[2] The dataset was generated randomly, but constrained to contain at least 1 example of each column of digits.

| Model | Strongly supervised / total traces | Accuracy for input length | |
|---|---|---|---|
| | | 500 | 1000 |
| NPI (Reed & De Freitas, 2016)[3] | 160 / 160 | <100% | <100% |
| NPL (Li et al., 2017)[3] | 10 / 160 | **100%** | <100% |
| PHP | 3 / 160 | **100%** | **100%** |

Table 2: Empirical results for the long-hand addition task. All models were trained with 16 traces per input length between 1 and 10, for a total of 160 traces, some of which strongly supervised.

sample complexity, EG can train PHPs which generalize to length-1000 inputs with 100% empirical test accuracy.

Moreover, we successfully learn models with as few as 3 strongly supervised demonstrations, compared to the 10 used by Li et al. (2017). However, we found that when the number of strongly supervised demonstrations was smaller than 10, early stopping of the training of the top-level policy was needed to learn a correct model. To obtain our reported results, we evaluated different snapshots of the model generated dur reporteding training.

## 6 DISCUSSION

In this paper we introduced the Parametrized Hierarchical Procedures (PHP) model for hierarchical representation of neural programs. We proposed an Expectation–Gradient algorithm for training PHPs from a mixture of strongly and weakly supervised demonstrations of an algorithmic behavior, showed how to perform level-wise training of multi-level PHPs, and demonstrated the benefits of our approach on two benchmarks.

PHPs alleviate the sample complexity required to train policies with unstructured memory architectures, such as LSTMs, by imposing the structure of a call stack augmented with program counters. This structure may be limiting in that it requires the agent to also rely on observable information that could otherwise be memorized, such as the building specifications in the NanoCraft domain. The benchmarks used so far in the field of neural programming are simple enough and observable enough to be solvable by PHPs, however we note that more complicated and less observable domains may require more expressive memory structures, such as passing arguments to sub-procedures. Future work will explore such structures, as well as new benchmarks to further challenge the community.

Our results suggest that adding weakly supervised demonstrations to the training set can improve performance at the task, but only when the strongly supervised demonstrations already get decent performance. Weak supervision could attract the optimization process to a different hierarchical structure than intended by the supervisor, and in such cases we found it necessary to limit the number of weakly supervised demonstrations, or weight them less than demonstrations annotated with the intended hierarchy.

An open question is whether the attractors strengthened by weak supervision are alternative but usable hierarchical structures, that are as accurate and interpretable as the supervisor's. Future work will explore the quality of solutions obtained by training from only weakly supervised demonstrations.

ACKNOWLEDGEMENTS

This research is supported in part by DHS Award HSHQDC-16-3-00083, NSF CISE Expeditions Award CCF-1139158 Berkeley DeepDrive, NSF Grant No. TWC-1409915, DARPA Grant No. FA8750-17-2-0091, NSF NRI Award 1734633, and gifts from Alibaba, Amazon Web Services, Ant Financial, CapitalOne, Ericsson, GE, Google, Huawei, Intel, IBM, Microsoft, Scotiabank, Splunk, VMware, Siemens, Cisco, Autodesk, Toyota Research, Samsung, Knapp, and Loccioni Inc.

Any opinions, findings, and conclusions or recommendations expressed in this material are those of the authors and do not necessarily reflect the views of the above organizations.

---

[3]Results provided by Li et al. (2017).

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

APPENDIX

## A    EXPECTATION–GRADIENT METHOD FOR PHPs

In weak supervision, only the observable trajectory $\xi = (o_{0:T}, a_{0:T-1})$ is available in a demonstration, and the sequence of memory states $\zeta = m_{0:T-1}$ is latent. This poses a challenge, since the space of possible memory trajectories $\zeta$ grows exponentially in the length of the demonstration, which at first seems to prohibit the computation of the log-likelihood gradient $\nabla_\theta \log \mathbb{P}(\xi|\pi_\theta)$, needed to maximize the log-likelihood via gradient ascent.

Our key insight is that the log-likelihood gradient can be computed precisely and efficiently using an instance of the *Expectation–Gradient (EG)* method (Salakhutdinov et al., 2003), which we detail below:

$$
\nabla_\theta \log \mathbb{P}(\xi|\theta) = \frac{1}{\mathbb{P}(\xi|\theta)} \nabla_\theta \mathbb{P}(\xi|\theta) = \sum_\zeta \frac{1}{\mathbb{P}(\xi|\theta)} \nabla_\theta \mathbb{P}(\zeta, \xi|\theta)
$$

$$
= \sum_\zeta \frac{\mathbb{P}(\zeta, \xi|\theta)}{\mathbb{P}(\xi|\theta)} \nabla_\theta \log \mathbb{P}(\zeta, \xi|\theta) = \mathbb{E}_{\zeta|\xi,\theta}[\nabla_\theta \log \mathbb{P}(\zeta, \xi|\theta)], \tag{1}
$$

where the first and third equations follow from the identity $\nabla_\theta x = x \nabla_\theta \log x$.

We start by assuming two-level PHPs, so that at any time $t$ the stack contains the root PHP and the PHP it invoked to select the elementary action. The stack also contains the program counters of these two PHPs, however we ignore the root counter to reduce complexity, and bring it back when we discuss multi-level hierarchies in Section 4.2.3 (and below).

Let us denote by $\eta_h^\tau(a_t|o_t)$ and $\psi_h^\tau(b_t|o_t)$, respectively, the stochastic operation and termination statements of procedure $h \in \mathcal{H} \cup \{\perp\}$, where $\perp$ is the root PHP. Let $(h_t, \tau_t)$ be the top stack frame when action $a_t$ is selected. Then the full likelihood $\mathbb{P}(\zeta, \xi|\theta)$ of the policy given an annotated demonstration is

$$
\mathbb{P}(\zeta, \xi|\theta) \propto \eta_\perp(h_0|o_0)\delta_{\tau_0=0} \prod_{t=0}^{T-1} \eta_{h_t}^{\tau_t}(a_t|o_t) \prod_{t=1}^{T-1} \mathbb{P}(h_t, \tau_t|h_{t-1}, \tau_{t-1}, o_t)\psi_{h_{T-1}}^{\tau_{T-1}}(1|o_T),
$$

where from the right-hand side we omitted the constant causal dynamics factor

$$
\mathbb{P}(o_{0:T}| \operatorname{do}(a_{0:T-1})) = \prod_{t=0}^{T} \mathbb{P}(o_t|o_{0:t-1}, a_{0:t-1}),
$$

and with

$$
\mathbb{P}(h_t, \tau_t|h_{t-1}, \tau_{t-1}, o_t) = \begin{cases} \psi_{h_{t-1}}^{\tau_{t-1}}(1|o_t)\eta_\perp(h_t|o_t) & \text{if } \tau_t = 0 \\ \psi_{h_{t-1}}^{\tau_{t-1}}(0|o_t)\delta_{h_t=h_{t-1}} & \text{if } \tau_t = \tau_{t-1} + 1. \end{cases}
$$

This formulation of the likelihood has the extremely useful property that $\nabla_\theta \log \mathbb{P}(\zeta, \xi|\theta)$ decomposes into a sum of gradients. To find the expected gradient, as in (1), we do not need to represent the entire posterior distribution $\mathbb{P}(\zeta|\xi, \theta)$, which would be intractable. Instead, we only need the marginal posteriors that correspond to the various terms, namely

$$
v_t(h, \tau) = \mathbb{P}(h_t{=}h, \tau_t{=}\tau|\xi, \theta)
$$
$$
w_t(h, \tau) = \mathbb{P}(h_t{=}h, \tau_t{=}\tau, \tau_{t+1}{=}\tau{+}1|\xi, \theta).
$$

With these, the EG trick gives us the gradient of the observable demonstration

$$\nabla_\theta \mathbb{P}(\xi|\theta) = \sum_{h \in \mathcal{H}} \sum_{t=0}^{T-1} \Bigg( v_t(h,0) \nabla_\theta \log \eta_\perp(h|o_t)$$

$$+ \sum_{\tau=0}^{t} \Bigg( v_t(h,\tau) \nabla_\theta \log \eta_h^\tau(a_t|o_t)$$

$$+ w_t(h,\tau) \nabla_\theta \log \psi_h^\tau(0|o_{t+1}))$$

$$+ (v_t(h,\tau) - w_t(h,\tau)) \nabla_\theta \log \psi_h^\tau(1|o_{t+1}) \Bigg) \Bigg). \qquad (2)$$

To allow the G-step (2), we take an E-step that calculates the marginal posteriors $v$ and $w$ with a forward–backward pass. We first compute the likelihood of a trajectory prefix

$$\phi_t(h,\tau) \propto \mathbb{P}(o_{0:t}, a_{0:t}, h_t{=}h, \tau_t{=}\tau),$$

up to the causal dynamics factor, via the forward recursion given by

$$\phi_0(h,0) = \eta_\perp(h|o_0),$$

and for $0 \leqslant t < T-1$

$$\phi_{t+1}(h',0) = \Bigg( \sum_{h \in \mathcal{H}, 0 \leqslant \tau \leqslant t} \phi_t(h,\tau) \eta_h^\tau(a_t|o_t) \psi_h^\tau(1|o_{t+1}) \Bigg) \eta_\perp(h'|o_{t+1})$$

$$\phi_{t+1}(h,\tau+1) = \phi_t(h,\tau) \eta_h^\tau(a_t|o_t) \psi_h^\tau(0|o_{t+1})).$$

We similarly compute the likelihood of a trajectory suffix

$$\omega_t(h,\tau) \propto \mathbb{P}(a_{t:T-1}, o_{t+1:T}|o_{0:t}, h_t{=}h, \tau_t{=}\tau),$$

via the backward recursion given by

$$\omega_{T-1}(h,\tau) = \eta_h^\tau(a_{T-1}|o_{T-1}) \psi_h^\tau(1|o_T),$$

and for $0 \leqslant t < T-1$

$$\omega_t(h,\tau) = \eta_h^\tau(a_t|o_t) \Bigg( \psi_h^\tau(1|o_{t+1}) \sum_{h' \in \mathcal{H}} \eta_\perp(h'|o_{t+1}) \omega_{t+1}(h',0) + \psi_h^\tau(0|o_{t+1})) \omega_{t+1}(h,\tau+1) \Bigg).$$

For efficiency considerations, note that this forward–backward graph has $(t+1)k$ nodes in layer $t$, where $k = |\mathcal{H}|$, but only $(t+1)k(k+1)$ edges to the next layer, rather than the naive $(t+1)(t+2)k^2$.

We can calculate our target likelihood using any $0 \leqslant t < T$, by taking

$$\mathbb{P}(\xi|\theta) = \sum_{h \in \mathcal{H}, 0 \leqslant \tau \leqslant t} \mathbb{P}(\xi, h_t{=}h, \tau_t{=}\tau) \propto \sum_{h \in \mathcal{H}, 0 \leqslant \tau \leqslant t} \phi_t(h,\tau) \omega_t(h,\tau),$$

so most efficient is to use $t = 0$

$$\mathbb{P}(\xi|\theta) = \sum_{h \in \mathcal{H}} \mathbb{P}(\xi, h_0{=}h, \tau_0{=}0) \propto \sum_{h \in \mathcal{H}} \phi_0(h,0) \omega_0(h,0).$$

Finally, the marginal posteriors are given by

$$v_t(h,\tau) = \frac{1}{\mathbb{P}(\xi|\theta)} \phi_t(h,\tau) \omega_t(\tau,h)$$

$$w_{T-1}(h,\tau) = 0,$$

and for $0 \leqslant t < T-1$

$$w_t(h,\tau) = \frac{1}{\mathbb{P}(\xi|\theta)} \phi_t(h,\tau) \eta_h^\tau(a_t|o_t) \psi_h^\tau(0|o_{t+1})) \omega_{t+1}(h,\tau+1).$$

As mentioned in Section 4.2.3, level-wise training of multi-level PHPs requires abstraction from lower levels and separation from higher levels. The former is achieved by rewriting weakly supervised demonstrations to show level-$i$ operations as elementary, for the purpose of training the next-higher level $i - 1$.

After level $i$ is trained, the level-$i$ PHPs that generated the demonstrations are decoded using the trained parameters. In our current experiments we used latent trajectories sampled from the posterior distribution, given by

$$\mathbb{P}(\zeta|\xi,\theta) = v_0(h_0,\tau_0) \prod_{t=0}^{T-2} \frac{\mathbb{P}(h_t,\tau_t,h_{t+1},\tau_{t+1}|\xi,\theta)}{v_t(h_t,\tau_t)},$$

where for each step $0 \leqslant t < T - 1$

$$\mathbb{P}(h_t,\tau_t,h_{t+1},0|\xi,\theta) = \frac{1}{\mathbb{P}(\xi|\theta)}\phi_t(h_t,\tau_t)\eta_{h_t}^{\tau_t}(a_t|o_t)\psi_{h_t}^{\tau_t}(o_{t+1})\eta_{\perp}(h_{t+1}|o_{t+1})\omega_{t+1}(h_{t+1},0)$$

$$\mathbb{P}(h_t,\tau_t,h_{t+1},\tau_t+1|\xi,\theta) = \delta_{h_{t+1}=h_t}w_t(h_t,\tau_t).$$

