# OpenReview forum: "Parametrized Hierarchical Procedures for Neural Programming"
_ICLR.cc/2018/Conference — Accept (Poster)_

### Official Review · AnonReviewer1 · 2017-11-26
**a good paper**

**Rating:** 6
**Confidence:** 1

**Review:**

In the paper titled "Parameterized Hierarchical Procedures for Neural Programming", the authors proposed "Parametrized Hierarchical Procedure (PHP)", which is a representation of a hierarchical procedure by differentiable parametrization. Each PHP is represented with two multi-layer perceptrons with ReLU activation, one for its operation statement and one for its termination statement. With two benchmark tasks (NanoCraft and long-hand addition), the authors demonstrated that PHPs are able to learn neural programs accurately from smaller amounts of strong/weak supervision.

Overall the paper is well-written with clear logic and accurate narratives. The methodology within the paper appears to be reasonable to me. Because this is not my research area, I cannot judge its technical contribution.

---

> ### Author Response · Authors · 2017-12-26
> **Re: a good paper**
>
> Thank you for your time and for your assessment. We are very excited about these results and are making updates to improve the paper.

---

### Official Review · AnonReviewer2 · 2017-11-27
**NPI with less supervision**

**Rating:** 6
**Confidence:** 3

**Review:**

I thank the authors for their updates and clarifications.  I stand by my original review and score.  I think their method and their evaluation has some major weaknesses, but I think that it still provides a good baseline to force work in this space towards tasks which can not be solved by simpler models like this.  So while I'm not super excited about the paper I think it is above the accept threshold.
--------------------------------------------------------------------------
This paper extends an existing thread of neural computation research focused on learning resuable subprocedures (or options in RL-speak).  Instead of simply input and output examples, as in most of the work in neural computation, they follow in the vein of the Neural Programmer-Interpreter (Reed and de Freitas, 2016) and Li et. al., 2017, where the supervision contains the full sequence of elementary actions in the domain for all samples, and some samples also contain the hierarchy of subprocedure calls.

The main focus of their work is learning from fewer fully annotated samples than prior work.  They introduce two new ideas in order to enable this:
1.  They limit the memory state of each level in the program heirarchy to simply a counter indicating the number of elementary actions/subprocedure calls taken so far (rather than a full RNN embedded hidden/cell state as in prior work).  They also limit the subprocedures such that they do not accept any arguments.
2.  By considering this very limited set of possible hidden states, they can compute the gradients using a dynamic program that seems to be more accurate than the approximate dynamic program used in Li et. al., 2017.

The main limitation of the work is this extremely limited memory state, and the lack of arguments.  Without arguments, everything that parameterizes the subprocedures must be in the visible world state.  In both of their domains, this is true, but this places a significant limitation on the algorithms which can be modeled with this technique.  Furthermore, the limited memory state means that the only way a subprocedure can remember anything about the current observation is to call a different subprocedure.  Again, their two evalation tasks fit into this paradigm, but this places very significant limitations on the set of applicable domains.  I would have like to see more discussion on how constraining these limitations would be in practice.  For example, it seems it would be impossible for this architecture to perform the Nanocraft task if the parameters of the task (width, height, etc.) were only provided in the first observation, rather than every observation.

None-the-less I think this work is an important step in our understanding of the learning dynamics for neural programs.  In particular, while the RNN hidden state memory used by the prior work enables the learning of more complicted programs *in theory*, this has not been shown in practice. So, it's possible that all the prior work is doing is learning to approixmate a much simpler architecture of this form.  Specifically, I think this work can act as a great base-line by forcing future work to focus on domains which cannot be easily solved by a simpler architecture of this form.  This limitation will also force the community to think about which tasks require a more complicated form of memory, and which can be solved with a very simple memory of this form.


I also have the following additional concerns about the paper:

1.  I found the current explanation of the algorithm to be very difficult to understand.  It's extremely difficult to understand the core method without reading the appendix, and even with the appendix I found the explanation of the level-by-level decomposition to be too terse.

2.  It's not clear how their gradient approximation compares to the technique used by Li et. al.  They obviously get better results on the addition and Nanocraft domains, but I would have liked a more clear explanation and/or some experiments providing insights into what enables these improvements (or at least an admission by the authors that they don't really understand what enabled the performance improvements).

---

> ### Author Response · Authors · 2017-12-26
> **Re: NPI with less supervision**
>
> Thank you for this valuable and detailed feedback.
>
> You are correct in pointing out that PHPs impose a constraining memory structure, and we added to Sections 1 and 6 notes on their limitations. In principle, any finite memory structure can be implemented with sufficiently many PHPs, by having a distinct procedure for each memory state. Specifically in NanoCraft, PHPs can remember task parameters by calling a distinct sub-procedure for each building location and size. This lacks generalization, which was also not shown for NanoCraft by Li et al. (2017). We expect the generalization achieved by limiting the number of procedures to be further enhanced by allowing them to depend on a program counter.
>
> This paper thus makes an important first step towards neural programming with structural constraints that are both useful as an inductive bias that improves sample complexity, and computationally tractable. We agree that more expressive structures will be needed as the field moves beyond the current simple benchmarks, which we hope this work promotes. We agree that passing arguments to hierarchical procedures is an important extension to explore in future work.
>
> We clarified in Section 4.2 and in the Appendix the explanations of the algorithm and of the level-wise training procedure. Specifically, in Section 4.2 we elaborated on the structure of the full likelihood P(zeta, xi | theta) as a product of the relevant PHP operations, and how this leads to the given gradient expression; and clarified the expression for sampling from the posterior P(zeta | xi, theta) in level-wise training.
>
> We added in Section 2 a short comparison of our method to that of Li et al. (2017). The main difference is that their method computes approximate gradients by averaging selectively over computation paths, whereas our method computes exact gradients using dynamic programming, enabled by having small discrete latent variables in each time step.

---

> > ### Comment · AnonReviewer2 · 2018-01-02
> > **Re: Re: NPI with less supervision**
> >
> > Thanks for your response.  The clarifications to Section 4.2 make the level-wise training algorithm more clear.
> >
> > The additional information in section 2 makes it clear how the gradient compuation differs, but it does not clarify where the gains come from.  Specifically, from the current results, it's not clear whether the gains come from (1) the ability to compute exact gradients rather than the approximate gradient computation used by Li et. al, or (2) the simpler PHP model is just easier to optimize in general, so it will work better regardless of the technique used for gradient computation.

---

> > > ### Author Response · Authors · 2018-01-05
> > > **Re:{^3} NPI with less supervision**
> > >
> > > Thank you for making this excellent point.
> > >
> > > Our experiments indicate that the gains of PHP are due to both (1) the ability to compute exact gradients for weakly supervised demonstrations (via the EG algorithm), and (2) the PHP model being easier to optimize than the NPI model. We added this observation to Section 5.1.
> > >
> > > As evidence for (2), consider the case of strongly supervised demonstrations, where NPL coincides with NPI and takes exact gradients. As shown in Figure 2 (blue curves at the 64 mark), with 64 strongly supervised demonstrations in the NanoCraft domain, the accuracy is 1.0 for PHP; 0.724 for NPL/NPI. In this case, PHP has lower sample complexity with comparable optimization algorithms, suggesting that this domain is solvable with a PHP model of lower complexity than the NPI model. We note, however, that Li et al. (2017) used batch size 1, whereas we used full batches and made no attempt to optimize the batch size.
> > >
> > > As evidence for (1), consider the case where 48 of the 64 demonstrations are weakly supervised (Figure 2, blue curves at the 16 mark). Here the success rate is 0.969 for PHP; 0.502 for NPL. Compared to the strongly supervised case above, this 70% increase in the gain of PHP over NPL is likely due to the exact gradients used to optimize the PHP model, in contrast to the approximate gradients of NPL.
> > >
> > > We are excited to present these results as they suggest a number of new research question, such as the effect of optimizing PHP with stochastic gradients, and we thank the reviewer for inspiring this direction for future research.

---

> > > > ### Comment · AnonReviewer2 · 2018-01-05
> > > > **Re:{^4} NPI with less supervision**
> > > >
> > > > I agree with your evidence for point (2).  However I don't see how your evidence for point (1) disambiguates between the two causes.  Couldn't it just as well be the case that the 70% increase of PHP over NPL is due to the fact that PHP is using a simpler model that is easier to optimize?

---

> > > > > ### Author Response · Authors · 2018-01-05
> > > > > **Re:{^5} NPI with less supervision**
> > > > >
> > > > > We agree that the evidence for (1) does not disambiguate the two causes well enough to support (1) with high confidence, and updated the paper to make this even clearer.
> > > > >
> > > > > We would also like to clarify the contributions of the paper:
> > > > > 1) We introduce the PHP model, which is simpler to optimize than the NPI model. That is, for a given dataset, the PHP model induces an optimization landscape in which a good solution is easier to find.
> > > > > 2) We propose an EG algorithm that computes exact gradients in this optimization landscape, allowing efficient optimization.
> > > > > 3) We show empirically that our model and algorithm outperform baseline models and algorithms.
> > > > >
> > > > > This work does not show that the approximate gradients of NPL are worse than exact gradients in optimizing the NPI model (with weak supervision), although this may well be the case when the NPL execution-path grouping loses useful path information. This work also does not show that the exact gradients of the full-batch EG algorithm are always better than approximate gradients; in fact, using SGD via minibatch EG may well be better (see e.g. [Keskar et al. 2017]). Finally, this work does not fully tease apart whether the gains are due to the PHP model inducing simpler optimization landscapes, or due to the EG algorithm utilizing them better, or both, although there is evidence that the PHP model is easier to optimize.
> > > > >
> > > > > These are all exciting research questions, and we thank the reviewer again for raising them. We believe that the simple and interpretable PHP model, the useful EG method, and the compelling empirical results presented here would be valuable to the community as a stepping stone towards such future research.
> > > > >
> > > > >
> > > > > [Keskar et al. 2017] On Large-Batch Training for Deep Learning: Generalization Gap and Sharp Minima, ICLR 2017

---

### Official Review · AnonReviewer3 · 2017-11-28
**.**

**Rating:** 6
**Confidence:** 2

**Review:**

Summary of paper: The goal of this paper is to be able to construct programs given data consisting of program input and program output pairs. Previous works by Reed & Freitas (2016) (using the paper's references) and Cai et al. (2017) used fully supervised trace data. Li et al. (2017) used a mixture of fully supervised and weakly supervised trace data. The supervision helps with discovering the hierarchical structure in the program which helps generalization to other program inputs. The method is heavily based on the "Discovery of Deep Options" (DDO) algorithm by Fox et al. (2017).

---

Quality: The experiments are chosen to compare the method that the paper is proposing directly with the method from Li et al. (2017).
Clarity: The connection between learning a POMDP policy and program induction could be made more explicit. In particular, section 3 describes the problem statement but in terms of learning a POMDP policy. The only sentence with some connection to learning programs is the first one.
Originality: This line of work is very recent (as far as I know), with Li et al. (2017) being the other work tackling program learning from a mixture of supervised and weakly supervised program trace data.
Significance: The problem that the paper is solving is significant. The paper makes good progress in demonstrating this on toy tasks.

---

Some questions/comments:
- Is the Expectation-Gradient trick also known as the reinforce/score function trick?
- This paper could benefit from being rewritten so that it is in one language instead of mixing POMDP language used by Fox et al. (2017) and program learning language. It is not exactly clear, for example, how are memory states m_t and states s_t related to the program traces.
- It would be nice if the experiments in Figure 2 could compare PHP and NPL on exactly the same total number of demonstrations.

---

Summary: The problem under consideration is important and experiments suggest good progress. However, the clarity of the paper could be made better by making the connection between POMDPs and program learning more explicit or if the algorithm was introduced with one language.

---

> ### Author Response · Authors · 2017-12-26
> **Re: your review**
>
> Thank you for these constructive comments.
>
> We added to Section 3 clarification of the connection between the POMDP formulation and program learning. In particular, the state s_t of the POMDP models the configuration of the computer (e.g., the tapes and heads of a Turing Machine, or the RAM of a register machine), whereas the memory m_t of the agent models the internal state of the machine itself (e.g. the state of a Turing Machine's Finite State Machine, or the registers of a register machine).
>
> The Expectation–Gradient method is somewhat similar to but distinct from the REINFORCE trick, which uses the so-called “log-gradient” identity \nabla_\theta{p_\theta(x)} = p(x) \nabla_\theta{\log p(x)} to compute \nabla_\theta{E_p[f(x)]}. In fact, we use that same identity twice to compute \nabla_\theta{\log P(\xi | \theta)}: once to express the gradient of log P(xi | theta) using the gradient of P(xi | theta); then after introducing the sum over zeta, we use the identity again in the other direction to express this using the gradient of log P(zeta, xi | theta).
>
> We added to Section 5.1 clarification that we did use the same total number of demonstrations for PHP as was used for NPL. The results for 64 demonstrations are shown in Figure 2, and the results for PHP with 128 and 256 demonstrations were essentially the same as with 64, and were omitted for figure clarity.

---

### Author Response · Authors · 2018-01-05
**Changes from original submission**

This is an overview of the main changes we introduced in revisions of the paper:

- In Sections 1 and 2.1, we removed the somewhat irrelevant discussion of the challenges of general RNNs.
- We reorganized Table 1.
- Based on feedback from Reviewer #3, in Section 3 we clarified the connection between the POMDP formulation and program learning.
- Based on feedback from Reviewers #2 and #3, in Section 4.1.1 we clarified the connection between the call stack and the agent memory, and noted basic properties of the PHP model complexity.
- Based on feedback from Reviewer #2, in Section 4.2.2 (formerly 4.2) we clarified the EG algorithm; in Section 4.2.3 (formerly 4.2.1) we clarified the level-wise training algorithm and updated the notation of layer indexes for consistency.
- Based on feedback from Reviewer #2, in Section 5.1 (Results) we addressed the causes of PHP gains; we removed the discussion of weak supervision which is repeated in Section 6.
- We removed Figure 3, which was somewhat redundant with Figure 1.
- Based on feedback from Reviewer #2, in Section 6 we addressed limitations of the PHP model and the need for more complicated benchmarks in this field.
- We made multiple minor clarifications and style improvements.

---

### Decision · Program_Chairs · 2018-01-29
**ICLR 2018 Conference Acceptance Decision**

**Decision:**

Accept (Poster)

**Comment:**

This paper is somewhat incremental on recent prior work in a hot area; it has some weaknesses but does move the needle somewhat on these problems.